# Impacts of Nanobubbles in Pore Water on Heavy Metal Pollutant Release from Contaminated Soil Columns

**DOI:** 10.3390/nano13101671

**Published:** 2023-05-18

**Authors:** Yihan Zhang, Zimu Song, Kosuke Sugita, Shan Xue, Wen Zhang

**Affiliations:** 1Department of Civil and Environmental Engineering, New Jersey Institute of Technology, Newark, NJ 07102, USA; yz27@njit.edu (Y.Z.); zs6167575@gmail.com (Z.S.); sx59@njit.edu (S.X.); 2Department of Mathematical Sciences, New Jersey Institute of Technology, Newark, NJ 07102, USA; ks823@njit.edu

**Keywords:** nanobubble water, heavy metal, soil column, soil fluidization, pollutant leaching

## Abstract

This study investigated the release of heavy metals from polluted soil under the pore water flow containing nanobubbles (NBs) to simulate natural ebullition. Three types of NBs (CH_4_, H_2_, and CO_2_) were generated in water and characterized, including bubble size, zeta potential, liquid density, and tension. The flow rate used in column tests was optimized to achieve proper soil fluidization and metal desorption or release. The leachate chemistries were monitored to assess the effect of NBs on conductivity, pH, oxidation–reduction potential (ORP), and dissolved oxygen (DO). The results showed that NBs in the pore water flow were significantly more effective in releasing Pb compared to DI water, with CO_2_ NB water being the most effective and H_2_ NB water being the least effective. CO_2_ NB water was also used to rinse column soil contaminated with four different metals (Pb, Cu, Zn, and Cr), which exhibited different leaching kinetics. Moreover, a convective–dispersion–deposition equation (CDDE) model accurately simulated the leaching kinetics and explained the effects of NBs on the key parameters, such as the deposition rate coefficient (*K_d_*), that affect the released metal transport. The findings could provide new insights into soil pollutant release under ebullition and soil remediation using water wash containing NBs.

## 1. Introduction

Due to past industrial and agricultural activities, numerous urban rivers, estuaries, and watersheds have become polluted with various contaminants, including synthetic organic compounds and heavy metals. Because of their hydrophobicity and recalcitrance to biodegradation, many organic pollutants tend to persist and accumulate in soils, sediments, and suspended solids, as well as in biological matrices [1]. Heavy metals could also bind to sediment matrices due to the porous nature and absorptive ability of sediment or soil organic matter. The sediment pollution has persisted over time and poses a significant challenge for the restoration of these water bodies [2,3]. The Environmental Protection Agency (US, EPA) has estimated that there are 1.2 billion cubic meters of contaminated surficial sediments in the US that could pose a risk to surface water resources, pollutant transport, public exposure, and even the security of drinking water supply [4,5]. With the implementation of the Clean Water Act (CWA), contaminated sediments have converted from being a sink to a source and thus deserve remediation [6]. 

Natural ebullition is a result of digenetic processes from certain aquatic sediments [7], which affects sediment contaminant transport [8,9]. Fine bubble formation in the sediment could be attributed to methanogenesis and fermentation, which produces gases such as CO_2_, CH_4_, and H_2_ [10]. The natural ebullition process creates a unique multi-phase flow phenomenon that affects sediment structures and sediment pollutant migration (e.g., desorption and release) [11]. The pore transport and rise could open up pore sizes and desorb contaminants in sediment. The release of sediment metals could be ebullition-facilitated [3]. For instance, arsenic (As) is released from soil into the aqueous solution under reductive conditions, whereas cadmium (Cd) becomes soluble under oxidative environments. Ebullition-induced mercury release from sediments has been observed in various aquatic environments [12], including lakes, reservoirs, and rivers [13]. Ebullition can also facilitate the release of lead and zinc from mining-contaminated sediments in rivers and streams [14]. Thus, the formation of microbubbles or nanobubbles in sediment may shift the local soil redox levels, altering the solubility and speciation of different heavy metals [15,16]. Moreover, CO_2_ NBs in water can suppress the pH down to 5.6 and may significantly increase the mobility of metal colloids or metal ions in the soil [17]. 

However, the bubble formation and blowout processes through sediment pores have not been sufficiently investigated. Many previous studies largely focused on sediment contaminant transport within the pore water; however, there is a limited understanding of the fine bubble transport characteristics and the associated impacts on sediment contaminant transport or migration [8]. Based on the existing understanding, the introduction of fine bubbles can create pressure gradients that can cause the transport of pollutants in soil to increase significantly [18]. The bubbles can increase the rate of diffusion of pollutants from the soil to the gas phase, which can then lead to atmospheric transport [19]. Additionally, the bubbles can create pathways for the movement of pollutants, increasing the likelihood of contamination of groundwater resources [20]. McLinn et al. reported that ebullition was employed to transfer coal tar from a synthetic polluted sediment through a sand cap under laboratory settings, which repeated the observed field transport phenomena reported by (US EPA, 2006) [21]. Exploring the effect of gas ebullition on the contaminant kinetics will provide an insight into the transport mechanism that governs pollutant release and resuspension. 

Based on the above-mentioned knowledge gaps and challenges, the present study evaluated the release of heavy metals from polluted sediment when the pore water flow was filled with NBs to mimic the natural ebullition. Different NBs (CO_2_, CH_4_, and H_2_) were generated in water and characterized to examine the influences of the NB water flow on the soil chemical properties, structural expansion, and metal leaching from spiked soil to the elute water phase. Among different typical sediment pollutants, heavy metals (e.g., Cu Fe, Pb, and Zn) were selected as model pollutants. The heavy metal release indicated by the calculated heavy metal fluxes (µg·L^−1^·min^−1^) was measured under different conditions. Finally, leaching kinetics’ modeling analysis was conducted to reveal the impacts of NBs on key parameters related to the transport behavior. Understanding and quantifying the influence of the fine bubble flows on sediment contaminant fate and transport could provide new insights into the effects of natural ebullition on soil pollutant fate and soil pollution remediation and prevention [19].

## 2. Materials and Methods

### 2.1. Preparation of Heavy Metal Contaminated Soil 

Miracle Gro Garden Soil was obtained from the Rutgers Soil Testing Laboratory, North Brunswick Township, NJ, USA and stored in precleaned brown glass bottles sealed with covers to avoid light and evaporation at 4 °C. These pristine soil samples were characterized and certified to be free of any known pollutants [22]. The major soil properties are provided in Appendix A in the Appendix A. To prepare for the experiments, the soil samples were dried in the oven at 104 ± 1 °C for 8 h to obtain a constant weight and sieved with a pore size of 2 mm. The contaminated soil was prepared to reach the reported heavy metal levels for polluted sediment in New Jersey and New York states [23]. Briefly, contaminated soil with 30 mg-Pb∙kg^−1^ was prepared by adding 30 mL of the 226 mg∙L^−1^ PbCl_2_ solution to 200 g air dry soil, which was mixed for 3 h with a Compact Digital Mixer (UX-50006-01, Cole-Parmer, Vernon Hills, IL, USA) and left in the atmosphere until the methanol completely evaporated. Similarly, another group of contaminated soil samples was prepared by adding the same volume of the four heavy metal solutions with the same molar concentration of 2 mmol∙L^−1^. Correspondingly, the soil samples contained 164 mg∙kg^−1^ of Pb, 128 mg∙kg^−1^ of Cu, and 130 mg∙kg^−1^ of Zn, and 104 mg∙kg^−1^ was prepared with solutions of PbCl_2_, Cu(NO_3_)_2_, Zn(NO_3_)_2_, and CrCl_3_ [24]. To confirm the contents of heavy metals in the spiked soil samples, these samples were digested by nitric acid (60–70% Trace Metal grade) at 85–95 °C to measure the leached metal concentration using ICP-MS, as detailed in Appendix A.

### 2.2. Generation and Characterization of NBs

NBs were produced in deionized (DI) water as suspensions following our reported membrane bubbling method [25]. As illustrated in Appendix A**,** compressed gases (i.e., CO_2_, CH_4_, and H_2_) were injected under a pressure of 414 kPa and a gas flow of 0.45 L·m^−1^ into a ceramic tubular membrane (model WFA0.1-Refractron, Newark, NY, USA). The air bubbles were dispensed into the flowing water that was pumped to circulate between the membrane module and the reservoir tank for 60 min to reach the maximum stable bubble concentration near 1.5 × 10^8^·mL^−1^ in 3-L water. The density of different NB saturated water was determined by measuring the weight of the aqueous solution with the same volume (20 mL). The temperature of the bubble suspension was also monitored at the same time [12]. The bubble size distribution and zeta potential of the water suspension of NBs were measured immediately after preparation using dynamic light scattering (DLS) on a Zetasizer Nano ZS instrument (Malvern Instruments, Malvern, UK). Each result was obtained from the average of five measurements. Furthermore, the mean concentration of NBs was determined using a nanoparticle tracking analysis (NTA) instrument (ViewSizer 3000, Horiba, Kyoto, Japan). The NTA graphs presented the standard deviations of five different measurements for each sample as error bars. A stable bubble size distribution and concentration (approximately 1.5 × 10^8^·mL^−1^) in water were obtained and were ready for use in the following experiments. The surface tension of all the NB water mentioned above was evaluated using the pendant drop method [26].

### 2.3. Column Setup and Characterization under a Bubble Water Flow

Column tests were conducted in a Perspex cylinder (Figure 1) with a 3 cm inner diameter and 30 cm inner height [27]. Each column was filled with 37 g of glass beads of 0.6 cm in diameter from the bottom up to form a 5-cm bottom layer as a support base and then with the 50 g contaminated soil to form the 5 cm soil layer that corresponds to a packing density of 1.40 g·cm^−3^, as suggested elsewhere [28]. On top of the soil layer, there was a 10 cm layer of glass beads to prevent soil swelling. The effluent exited the column 10 cm above the topsoil surface. To mimic the natural pore water flows or gas ebullition [29,30,31,32], different flow rates (5–20 mL·min^−1^) or fluxes from 0.71 to 2.83 mL·cm^−2^·min^−1^ were injected into the soil column from the bottom using a peristaltic pump (Masterflex Model 77200-62, Cole-Parmer Instrument Company, Vernon Hills, IL, USA). The upflow water flow was expected to result in a moderate fluidization level without causing any turbulence or soil mixing. 

The bulk density and porosity of the packed column were measured gravimetrically [33]. The porosity was calculated as the difference between the bulk density and the packing density of the soil [34]. Additional to the soil structures, the eluate was collected to measure turbidity (Ratio XR Turbidimeter model 43900, HACH, Loveland, CO, USA), conductivity (conductivity sensor, PS-3210, PASCO, Portland, OR, USA), pH, and ORP using PASCO Xplorer GLX sensors (PASCO, Roseville, CA, USA).

### 2.4. Evaluation of the Metal Release from the Spiked Soil under a Bubble Water Flow

To evaluate the effect of different NBs on Pb release, the different types of bubble water (CO_2_, CH_4_, and H_2_) were pumped into the soil column at a fixed flow flux of 0.71 mL·min^−1^·cm^−2^. The Pb-spiked soil columns underwent two parallel soil-washing tests. The column was sealed by a rubber plug at the top to keep the drainage from being exposed to air. Effluent that was not exposed to air was collected from a sidewall port 20 cm above the bottom of the cylinder. Then, 10 mL of the effluent samples was taken directly from the effluent collection point every 10 min for the first 60 min and every 30 min for the following 60 min to analyze the leached heavy metal concentration. Deionized (DI) water without NBs was pumped into the column at the same flux for comparison as a negative control group. For the soil spiked with multiple metals (i.e., Pb, Cu, Zn, and Cr), CO_2_ NB water was used to rinse the column soil and assess the leaching efficiencies. To determine significant differences between the control and experimental data, one-way analysis of variance (*t*-test, two-sided, with a significance level of *α* = 0.05) was employed to assess the data statistics of heavy metal concentrations in the leachate. All statistical analysis and data plotting were carried out using Excel 2016 and Origin version 2020b [35]. All the standard deviations (error bars) were calculated from two parallel experiments’ data, which are subjected to uncertainties in sampling and instrument analysis.

### 2.5. Modeling Analysis of the Metal Leaching Kinetics 

The leached heavy metal concentration, *C(t)*, often varies with the elapsed or leaching time, *t* (min), in a first-order kinetic deposition term [36,37]. Recently, Zhang et al. developed a time fractional convective–dispersion–deposition equation (CDDE) model in Equation (1) [38], which described the tailing property of the pollutant leaching kinetics of different heavy metals such as Zn, Mn, Ag, and Ga from contaminated soil.
(1)$$R_{f}\frac{\partial^{\alpha }C}{\partial t^{\alpha }}=D\frac{\partial^{2}C}{\partial x^{2}}-v\frac{\partial C}{\partial x}-K_{d}C$$
where *R_f_* (dimensionless) is the retardation coefficient and *D* is the hydrodynamic dispersion coefficient, both of which are calculated in detail in Appendix A. In addition, *v* is the average linear water velocity (cm·min^−1^), *K_d_* is the deposition rate coefficient that describes the adhesion strength of target pollutants with soil media or surface, and 𝛼 is the time fractional derivative order (between 0 and 1), which reflects the retention effect of the soil column on pollutants. Generally, a strong tailing effect occurs to the leaching process of pollutants when *α* becomes small. This CDDE model in Equation (2) reduces to a CDE model when *α* = 1, as follows:(2)$$R_{f}\frac{\partial C}{\partial t}=D\frac{\partial^{2}C}{\partial x^{2}}-v\frac{\partial C}{\partial x}-K_{d}C$$

The above two models indicate five parameters may affect the transport of heavy metal ions in a porous soil column under washing, including the time fractional derivative order (*α*), retardation factor (*R_f_*), hydrodynamic dispersion coefficient (*D*), flow velocity (*v*), and deposition rate coefficient (*K_d_*). The above two models were employed to fit the experiment data of the leached concentrations of heavy metals under different rinsing conditions. Our assumption is that the presence of NBs could affect *α* or *K_d_*, which were used as the fitting parameters. The detailed solutions of the above CDDE or CDE model are provided in Appendix A.

## 3. Results and Discussions

### 3.1. Soil Characterization

Appendix A summarizes the major soil properties, such as the organic matter content of 3.0%, the bulk-density of 0.4 kg·L^−1^, soil porosity of 41%, and a pH of 5.76. The measured soil properties represent a sandy loam that is composed primarily of gravel and sand with very high organic-carbon content of 3.0%, which may strongly bind with Pb^2+^ and other heavy metals [39]. Soil organic matter plays an essential role in heavy metal adsorption (e.g., Pb and Cd) and sequestration as a result of the formation of metal-organic complexes [40,41]. 

### 3.2. Bubble Characterization 

Figure 2a shows the bubble sizes and zeta potentials of CH_4_, CO_2_, and H_2_ NBs in tap water. These NBs had hydrodynamic diameters from 300 to 500 nm and negative surface charges. H_2_ NBs had the greatest zeta potential of −38.2 mV, while CH_4_ NBs has the highest zeta potential of −5.85 mV. Figure 2b indicates that the water suspension densities for the three types of NBs are consistently lower than the water density (1 g·mL^−1^). According to the NTA measurement, the concentration of the original water suspension of these NBs was approximately 1.5 × 10^8^ #·mL^−1^. Clearly, the presence of NBs slightly reduced the density of the water suspension. The surface tension of NB water was almost the same for the three types of NBs. For example, the surface tension of CH_4_ NB water was shown to be 70.20 mN·m^−1^, which is slightly lower than the water surface tension (72 mN·m^−1^ at room temperature). Appendix A.

### 3.3. Effects of Nanobubble Water on Soil Structures and Soil Chemistries 

#### 3.3.1. Soil Fluidization

To compare the effects of the water flow with/without NBs on soil structures, the bubble water and DI water were purged separately in the soil column without placing the 10 cm glass bead layer on the top. Figure 3a shows the height of the expanded soil layer, which was consistently greater for the groups treated by the NB water compared to that by DI water under different flow rates. This result indicates that the presence of NBs could significantly increase the expansion and fluidization of the packed soil by 20–40% compared to the negligible expansion under DI water. Correspondingly, the calculated soil porosity in Figure 3b was shown to rise from 40% to 70% as the flow rate of NB water increased up to 2.12 mL·min^−1^·cm^−2^. At the flow rate of 2.83 mL·min^−1^·cm^−2^, there was a sudden “burst out” of the expanded soil column that ejected a vigorous jet of water and sediment into the overlying water and caused significant turbulent mixing, as illustrated in the inset photos of Figure 3b. Thus, for the following heavy metal leaching tests, a constant flow rate of 0.71 mL·min^−1^·cm^−2^ was chosen unless indicated otherwise. Appendix A.

#### 3.3.2. Water Chemistries of the Soil Elute

Figure 4a compares the elute conductivity under the purging flow of DI water and NB water, which resulted in significant differences of ionic species release. The DI water caused a high initial conductivity due to the rapid leaching of ionic species from the washed soil, which after approximately 20 min reached a plateau near zero. Though the flow rate of DI water did not vary the elute conductivity significantly, the elute conductivity appeared to increase slightly when a flow rate of DI water at 0.71 mL·min^−1^·cm^−2^ was used, probably because a longer water retention time permitted more effective leaching of ionic species from the soil. However, the wash process by different NB waters extended the leaching time significantly (e.g., from 20 min to nearly 60 min) besides the rapid leaching at the beginning. For example, CO_2_ NBs caused a consistently high conductivity in elute, indicating that CO_2_ NBs appeared to mobilize and leach out higher amounts of ionic species from the soil than DI water or other types of NBs. All NBs in soil water are expected to carry negative charges (between −5 mV and −35 mV) and, therefore, may attract positively charged cations from the soil, which explains the higher conductivity in the elute water from the NB water treatment groups compared to the DI water treatment group. 

Figure 4b–d compares the changes of the elute pH, ORP, and DO over the sampling time under the same flux of DI water or the NB water flux at 0.71 mL·min^−1^·cm^−2^. Figure 4b shows that CO_2_ NBs and DI water did not result in significant changes of the elute pH, whereas CH_4_ and H_2_ NBs both increased the elute pH appreciably to 7 and 6, respectively. This pH rise is likely due to the stripping of the dissolved CO_2_ from ambient air by CH_4_ and H_2_ NBs. Figure 4c shows that the elute ORP levels for CH_4_ and H_2_ NBs were progressively reduced from 200 mV to 150 mV or nearly 30 mV after 120 min, which partly results from the reduction of the dissolved oxygen (e.g., from 8 mg·L^−1^ to 6.5 mg·L^−1^), as shown in Figure 4d. Due to the linear relationship between ORP and the logarithm of oxygen concentration, an ORP of −96 mV in the aqueous phase is equivalent to a DO of 0.1 mg·L^−1^ [42]. In addition, the existence of CH_4_ in water can cause a micro-oxygen environment such that the ORP value is in the range from 0 to −470 mV [43]. The generation of hydrogen bubbles in water for 30 min can reduce the ORP to −115 mV, and after boiling for 15 min, in contrast, the oxidation–reduction potential is reduced to −79 mV [44]. However, despite the low DO caused by CO_2_ NB, ORP did not vary significantly with the rinsing time of CO_2_ NBs. According to Equation (S4), as CO_2_ NBs are purged into the soil column, the CO_2_ vapor pressure in the elute will increase, which will linearly increase ORP or E_H_ if pH remains constant; this in fact dropped slightly (red circle data) according to Figure 4c and thus further caused the increase of ORP.

### 3.4. Release of Soil Heavy Metals under the Soil Washing by Different NB Water 

Figure 5 shows that the release kinetics of the Pb concentrations from the soil was highly dependent on the types of NBs. The released Pb concentrations started to decline over time due to the elution or desorption from the spiked soil within the first 20 PVs. The leaching rate reduced significantly after 20 PVs. The effluent Pb concentrations tended to stabilize and reached a relatively low level after 30 PVs. The initial leached concentration of Pb under DI water wash is significantly lower than that under the NB water wash, suggesting an enhanced leaching process of heavy metal in soil by the presence of NB_s_.

Among different NBs, CO_2_ NB water led the highest or fastest rate of the Pb release (nearly 5.12 mg·L^−1^ at the time of 10 min) within the first 2 h (*p* < 0.05). By comparison, the DI water wash only led to an immediate concentration of 3.05 mg·L^−1^. H_2_ or CH_4_ NBs did not result in significant differences. Again, the enhanced Pb release under the pore flow water of NBs should be attributed to the negative surface charges of NBs that enabled electrostatic attractions towards positively charged Pb, among other possible mechanisms [25]. In particular, for CO_2_ NBs, the enhanced Pb release could also be attributed to the soil pH drop or slight acidification. Ionic Pb in soil may be absorbed by organic matter or coprecipitate with oxygen-containing groups such as carboxyl (-COOH) and hydroxyl (-OH) [45]. When the soil pH decreases, the binding strength between Pb and oxygen-containing anions will be reduced [46], which remarkably promotes the leaching process. According to Yang et al. [39], when the pH of the red soil was below 2, the desorption rate of Pb was found to be greater than 80%. However, the rate decreased to approximately 50% when the pH was between 2.9 and 3.4. 

Thus, we further compared the leached concentrations of Pb, Cu, Zn, and Cr using CO_2_ NB water in Figure 5b–e. The soil sample was prepared by adding the same volume of the four heavy metal solutions with the same molar concentration of 2 mmol∙L^−1^. Compared to DI water, the water rinse with CO_2_ NBs consistently caused higher leached concentrations of heavy metals, especially at the sampling time of 10 min. Moreover, the leached levels of Cu (288.55 µg·L^−1^) and Zn (2433 µg·L^−1^) were substantially higher than Pb (35.86 µg·L^−1^) and Cr (71.94 µg·L^−1^), which indicates that Cu and Zn leached faster than Pb and Cr under the same CO_2_ NB water wash. The different release rates of different heavy metals may result from their interactions with NBs, soil binding, and desorption characteristics under the pore water rinse. Finally, we calculated the total released Pb amount by integration of the time-dependent concentration and volume of elute, which suggested that only 3–8% of Pb was leached in the 2 h experiments. A higher or complete removal of the spiked heavy metals requires further optimization of operations, such as increasing the washing time, choosing the washing liquid/soil (L/S) ratio, and adding chemical surfactants or chelate, which are not the focuses of this study.

### 3.5. Leaching Kinetics Modeling Analysis 

The smooth curves of different lines in Figure 5 are the model fits generated by the CDDE model by varying the two fitting parameters (*α* and *K_d_*), which yielded an excellent match with experimental data points. Appendix A further compares the model fit and experimental data using the CDDE and CDE models. In the model fitting process with CDE, only one fitting parameter (*K_d_*) can be varied, whereas the CDDE model fit could generate the potential changes of the time fractional derivative (*α*) under different rinsing solutions. Clearly, both models achieved excellent fitting, with their fitting parameters summarized in the caption of Appendix A, which shows that the fitting values of *α* remain constant at 0.97 or 1 for the two models and *K_d_* reduced when NBs were present. In particular, the curve fit for the data of CO_2_ NBs resulted in a *K_d_* value of 0.046, much lower than that of DI water (0.0925), which suggests that NBs reduced the adhesion strength of Pb on the soil surface. Moreover, the model fitting results reveal that the retention effect of the soil column on Pb is low without a significant tailing effect, as 𝛼 is almost 1. Thus, after the majority of the pollutant has been transported or leached, a very small residual amount of pollutant remains trapped or bound within the soil matrix. By contrast, significant tailing was reported for the release of iron and mercury from the contaminated soils that were barren, alkaline, and deficient with organic matters [47]. The soil sample used in this study was rich in organic matter and had a pH of 5.76, which may explain the monitor tailing effect on the Pb release.

Appendix A show the simulation results using the CDDE and CDE models to evaluate the impacts of the four parameters, retardation factor (*R_f_*), hydrodynamic dispersion coefficient (*D*), flow velocity (*v*), and deposition rate coefficient (*K_d_*), on the leaching kinetics of Pb under the influences of three types of NBs and DI water. For instance, Appendix A shows the calculated concentration of leached Pb under elution with the CH_4_ NB water when varying *R_f_* from 1 to 10 with the two models, where α = 0.97 and *K_d_* = 0.07974 were used. Clearly, the CDE model shows that increasing *R_f_* led to a slower release or leaching of Pb due to the retardation effect, whereas the CDDE model simulation shows that Rf barely affects the leaching kinetics. Similarly, the model simulation in Appendix A shows that the leaching kinetics does not depend on the variations of *D* in the range (1.45 × 10^−5^ to 1.45 × 10^−3^ cm^2^·min^−1^). Likewise, the influence of *v* on the leaching kinetics of Pb is also negligible. By contrast, the impact of *K_d_* on the leaching process of Pb is obvious. Increasing *K_d_* from 0.04620 to 0.7974 min^−l^ also slowed down the release of Pb, as a higher deposition rate coefficient means that Pb tends to adhere to the soil and thus leach less efficiently.

Appendix A shows the simulation results using the CDDE model to evaluate the impact of the time fractional derivative (α). When α increases from 0.97 to 1.0, the leaching velocity slightly increases at a later stage after 10 PV. The time fractional derivative is used to account for the non-integer order of diffusion that occurs when solute moves through a porous medium. When α increases, the diffusion process will become more anomalous and therefore the leaching process is more sensitive to changes in the concentration of the solute in the liquid phase. In other words, the leaching process becomes more efficient and the rate of extraction of the solute from the solid material increases.

## 4. Conclusions

The presented study investigated the release of heavy metals from polluted sediment under the pore water flow containing NBs made with H_2_, CO_2_, and CH_4_. The sandy loam soil had a high organic-carbon content and a pH of 5.76. Three types of NBs (CO_2_, CH_4_, and H_2_) in the pore water caused soil expansion, as indicated by soil column height increase. A constant pore flow flux of 0.71 mL·min^−1^·cm^−2^ was chosen for proper soil fluidization and heavy metal leaching. The conductivity of leachate using DI water under different flow rates decreased rapidly and reached a plateau near zero after 20 min, while different NBs water extended the washing process and CO_2_ NBs caused a comparatively higher conductivity; CH_4_ and H_2_ NBs increased the elute pH to 7 and 6, respectively, while CO_2_ NBs decreased the elute pH to 5.5; ORP levels for CH_4_ NBs and H_2_ NBs were reduced from 200 mV to 150 mV or nearly 30 mV, while CO_2_ NBs increased the ORP slightly. All three types of NBs decreased the elute DO to 5.5–6.8 mg·L^−1^. The presence of NBs in water wash increased the Pb release by 57.15–198.77% compared to DI water, probably because the negative surface charges and high specific surface areas enabled electrostatic attraction of soil cations. In particular, CO_2_ NBs achieved the highest leaching rate of Pb, followed by CH_4_ and H_2_ NBs that led to similar leaching rates of Pb. The rinse of the soil column spiked with four different metals (Pb, Cu, Zn, and Cr) by CO_2_ NB water resulted in different leached concentrations of Cu (288.55 µg·L^−1^) and Zn (2.433 mg·L^−1^) at 10 min, which were much higher than Pb (35.86 µg·L^−1^) and Cr (71.94 µg·L^−1^). Clearly, the interactions of heavy metal cations and NBs, soil binding, and desorption characteristics may affect the release kinetics. To provide insights into the effects of NBs on soil pollutant transport behavior, a convective–dispersion–deposition equation (CDDE) and a convective–dispersion equation (CDE) model were employed to fit the leaching data to obtain key parameters such as retardation factor (*R_f_*), hydrodynamic dispersion coefficient (*D*), flow velocity (*v*), and deposition rate coefficient (*K_d_*). Results showed that leaching kinetics does not depend on the variations of *D* in the range 1.45 × 10^−5^ to 1.45 × 10^−3^ cm^2^·min^−1^ nor *v* in the range 0.71 to 2.84 mL·min^−1^·cm^−2^. The CDE model demonstrates that as Rf increases, the release of Pb slows down because of the retardation effect, while the CDDE model simulation indicates that the leaching kinetics of Pb is barely affected by *Rf*, while increasing *K_d_* from 0.04620 to 0.7974 min^−l^ also led to a slow and reduced release of Pb in both models.

## Figures and Tables

**Figure 1 nanomaterials-13-01671-f001:**
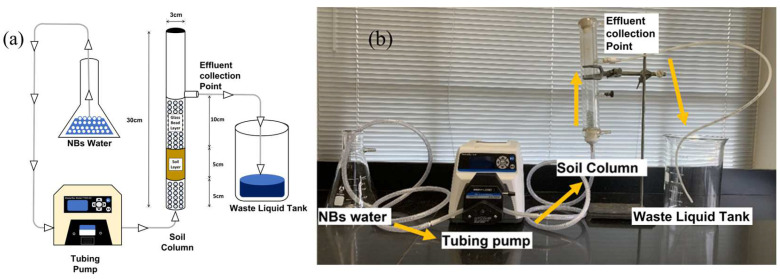
(**a**) Schematic of the soil column and upflow soil washing to leach heavy metals; (**b**) Photo of the soil column.

**Figure 2 nanomaterials-13-01671-f002:**
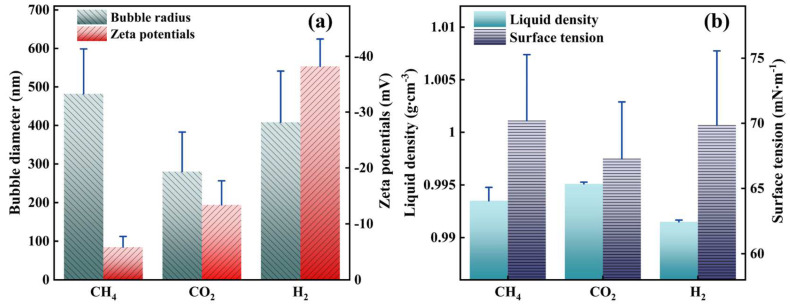
(**a**) The bubble diameter and zeta potentials in tap water; (**b**) Liquid density and surface tension of three different NBs water.

**Figure 3 nanomaterials-13-01671-f003:**
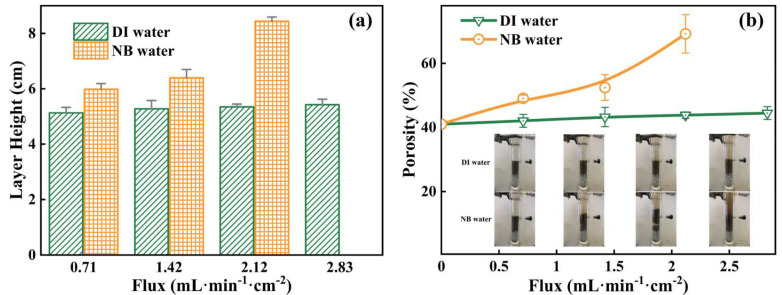
The effects of DI water and NBs water flows on soil structures: (**a**) Soil layer height; (**b**) Soil column porosity (the solid lines are a guide to the eye).

**Figure 4 nanomaterials-13-01671-f004:**
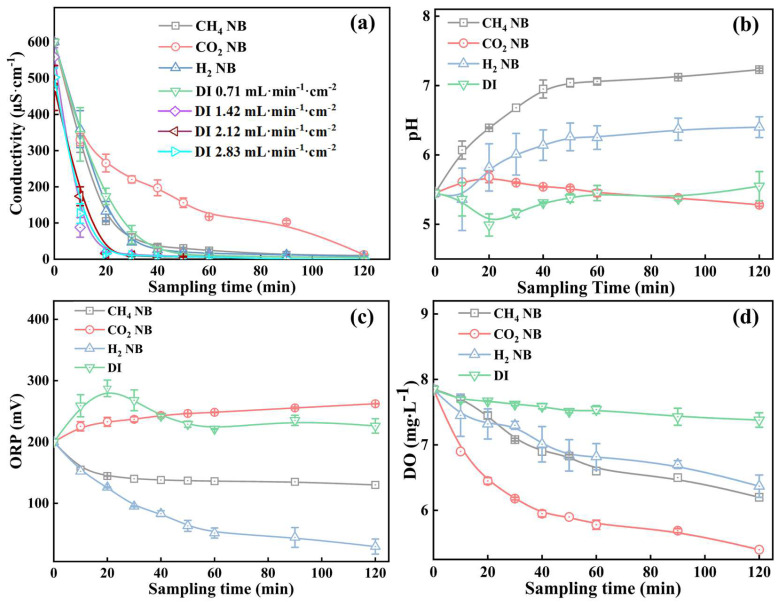
(**a**) Conductivity of elute using different flow rates of DI water or at the NB water flux of 0.71 mL·min^−1^·cm^−2^; (**b**–**d**) The elute pH, ORP, and DO changes under different purging time by the NB water.

**Figure 5 nanomaterials-13-01671-f005:**
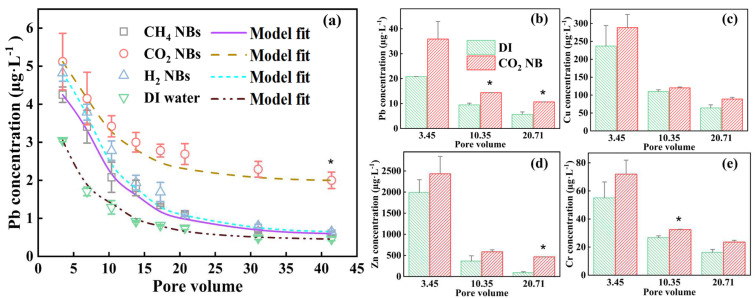
(**a**) Pb concentration in the effluent using DI water and three types of NB water under the same water flux of 0.71 mL·min^−1^·cm^−2^ . The data points correspond to the experimental measured Pb concentrations at different sampling times. The lines represent the model fits using the CDDE model in Equation (1). (**b**–**e**) The leached concentrations of four heavy metals (Pb, Cu, Zn, Cr) from the soil with mixed heavy metal spike using DI water and CO_2_ NB water; * indicates that the difference between the two treated groups is significant (*p* < 0.05). Pore volume (PV) = (*Q·t*)/(*V_T_·φ*), where the flow rate (*Q*) is 5 mL·min^−1^, *t* is the sampling time (min), the soil layer volume (*V_T_*) is 35.34 cm^3^, and the porosity (*φ*) is 0.41.

## Data Availability

The data presented in this study are available on request from the corresponding authors.

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
