# Peer review of "Impacts of Nanobubbles in Pore Water on Heavy Metal Pollutant Release from Contaminated Soil Columns"

_nanomaterials, 2023, doi:10.3390/nano13101671_

Round 1
Reviewer 1 Report
The manuscript by Zhang et al. reports the effect of the addition of nanobubbles in water on the efficiency of extraction of metal ions from contaminated soil. A model is applied to obtain useful parameters such as retardation factor, hydrodynamic dispersion coefficient, and deposition rate coefficient. The manuscript is well written and the figure quality is high. I have only one question, that the authors could address in their discussion. How is the effectiveness of the ion extraction change when surfactants/oil/contaminants are present in the contaminated oil? Minor changes required are below:
- Acronyms should be defined in the abstract (ORP, and DO) and the first time they appear in the text.
- Reference 26 is missing page and volume number
- Reference 26 is the previously published method used to create nanobubbles. Brief details of the process should be included in this manuscript for completeness. How long after preparation can the NB water be used and retain its properties?
- The origin of the error bars in each figure should be included in the caption, including details such as how it was calculated (st dev?), number of repeats and sources of error.
-
Reviewer 2 Report
This is an interesting study of the influence of nanobubbles on the extraction of heavy metals from polluted soil under pore water flow containing nanobubbles. Three filling gases for the nanobubbles and four different metals are investigated. The experimental arrangements used to investigate the soil and nanobubble properties are carefully described, as is the measurement of metal leaching as a function of time.
The discussion is generally clear. The difference in the effect of nanobubbles for Pb compared with Cu, Zn and Cr is striking. It appears that effects at short times may be masked (in terms of statistical significance) by the larger error bars for those measurements: it would be interesting to know whether there were practical issues which led to those larger scatters.
As regards the mathematical analysis the identification of the position variable x with the travel distance (v t) is not, as implied in the Supplement, exact. It implicitly assumes that dispersion is slow compared with advection. This is probably the case, but should be noted, and it means that, for example, equation S15 leads to a time-invariant concentration in the case v=0 -- that is, neither advection nor dispersion.
Points that require comment are:
1. The differences between the CDE and CDDE are marked, despite the fact that the difference between the differential order 0.97 in the CDDE and 1.0 leading to the CDA is very small. Is there a ready explanation of this?
2. The analytic results show that C(t) should tend exponentially to zero for the CDE, whereas the results in Figure S4 appear to reach a non-zero limit (for example, about 0.56 in Figure S4(a)). Why is this the case? What is the physical reason to expect a finite limiting concentration?
3. The expression D Kd/v2 should be dimensionless -- this is not the case for the values quoted in the Supplement. Also Kd appears as K d and Rf as R f in several places.
For the reader there is little advantage in changing the time scale from minutes to pore volume, and it would be more convenient to have a consistent scale.
The abbreviations ORP (oxidation-reduction potential) and DO(dissolved oxygen) should be defined.
This is an interesting study, meriting publication, but the authors should consider minor modifications to address the points raised above.
